# "Why take the patient back home?": Exploring the lived experiences of caregivers of COVID-19-infected individuals in Blantyre, Malawi

**Alinane Linda Nyondo-Mipando**[1]*, **Deborah Nyirenda**[2], **Leticia Suwedi-Kapesa**[1,3], **Marumbo Chirwa**[1], **Victor Mwapasa**[4]

**1** Department of Health Systems and Policy, School of Global and Public Health Kamuzu University Health Sciences, Blantyre, Malawi, **2** Malawi Liverpool Wellcome Trust, Blantyre, Malawi, **3** Public Health Institute of Malawi, Lilongwe, Malawi, **4** Department of Public Health, School of Global and Public Health Kamuzu University Health Sciences, Blantyre, Malawi

* lmipando@kuhes.ac.mw

## Abstract

The Corona Virus Disease 2019 (COVID-19) pandemic continues to have serious health and socio-economic consequences. In Malawi, COVID-19 cases are managed at home, with hospital admission reserved for severe cases. This study described the lived experiences of caregivers of COVID-19-infected individuals in Blantyre, Malawi. This descriptive qualitative study was conducted from January to June 2021 in Blantyre, Malawi, among caregivers of confirmed SARS-COV-2 cases enrolled in the SARS-CoV-2 study and aimed to explore infections, transmission dynamics, and household impact. We conducted 37 in-depth interviews with caregivers of SARS-COV-2 cases who were purposively sampled. We captured data using telephonic interviews, digitally recorded, transcribed verbatim, managed using NVivo, and analysed it using a thematic approach guided by the theory of care-giving dynamics. Caregivers stated that the economic status of a family largely influenced caregiving roles and abilities because it determined the resources that a household could access or not. Caregivers expressed being committed to their role despite being ill-prepared to manage a COVID-19 patient at home, in addition to fears about the contagious nature of COVID-19. They prioritised their patients' health by ensuring that they were present to offer nutritional and medical treatment. Caregivers highlighted challenges faced in the implementation of preventive measures because of financial limitations and cultural factors. They further expressed fear resulting from the increase in death rates, and the lack of proper information challenged their expectation of having their patients healed. Caregivers stated that they managed their role by sharing responsibilities, getting better at it with time, and getting support from religious institutions and social networks. Caring for confirmed cases of COVID-19 demanded commitment from the caregivers while ensuring that the transmission of the virus was minimised. There is a need to support households in isolation with the right information on how to manage their patients and streamline social support for the ultra-poor.

**Data Availability Statement:** The datasets used and/or analysed during the current study are all included in the manuscript as part of the results.

**Funding:** The study was funded by the National Institute for Health and Care Research (NIHR) under the Global Effort on COVID-19 (GECO) Health Research grant to Prof. Victor Mwapasa. Grant number: MR/V02860X/1. The funders had no role in study design, data collection and analysis, decision to publish, or preparation of the manuscript. The views expressed are those of the author(s) and not necessarily those of the NIHR or the Department of Health and Social Care.

**Competing interests:** The authors have declared that no competing interests exist.

## Introduction

Corona Virus (COVID-19) is a global pandemic caused by Severe Acute Respiratory Syndrome Corona Virus (SARS-CoV- 2). Most people diagnosed with COVID-19 usually present mild to moderate symptoms requiring home management and nursing. COVID-19 is a highly infectious disease, and the World Health Organisation (WHO) recommends several measures to prevent the spread of the pandemic. These measures include the mandatory wearing of masks, getting vaccinated against the disease, keeping a physical distance of a metre, cleaning hands frequently with soap and water or an alcohol-based hand rub, sneezing into a bent elbow, and self-isolation if one has symptoms or tests positive for COVID [1]. Since COVID-19 is a global pandemic, all countries, including Malawi, have adopted these preventive measures.

The first case of COVID-19 in Malawi was reported on April 2,2020 [2]. As of 29November 2022, 88,086 people tested positive for COVID-19 in Malawi, with a case fatality rate of 3.05% [2]. The second wave, from December 2020 to May 2021, had the highest case fatality rates among all the COVID-19 waves in Malawi. Malawi has a four-tiered health system that includes the community, primary, secondary, and tertiary levels [3]. Testing for COVID-19 infection in Malawi is done in selected primary healthcare facilities but is mostly done in secondary-level facilities. All admissions are at both secondary and tertiary-level facilities. Malawi followed a staggered approach in developing sites for testing and admission, with tertiary hospitals championing the services. Hospital admission in Malawi is reserved for severe cases with a critical disease, severe respiratory distress or hypoxemia, aggravation of co-morbidities, shock, and severe bacterial or viral co-infections, among others [4]. In contrast, Malawi National Guidelines state that mild and moderate cases of COVID-19 be managed at home, focusing on preventing transmission to others through isolation from family, pets, and the public, strict physical distancing, hand hygiene, cough etiquette, cleaning, and disinfection of frequently touched surfaces, and monitoring for deterioration. These caregivers are usually family members who offer the services voluntarily and may not be adequately prepared for the role [5].

In terms of quarantine, the Malawi Ministry of Health's National Guidelines for quarantine in the context of COVID-19 recommend that a person who had close contact with a confirmed case of COVID-19 within a period of 0 to 14 days or a traveller entering the country from other countries affected by the COVID-19 pandemic be quarantined for 14 days [4]. Home quarantine is the voluntary isolation of an exposed person in the home, including a fence separating the suspect from the rest of the family or community, and dedicated room bedding, towels, cutlery, cups, and plates only for their use for the duration of the 14 days [4]. The government is obliged to orient and provide educational material to persons in the homes of the exposed persons to ensure their understanding of IPC. The rooms that the exposed person uses should be well-ventilated and preferably self-contained, among other criteria, and if some of the criteria are not met, the guidelines recommend institutional quarantine. Ideal sites for quarantine are hotels, school dormitories, institutional hostels, and other facilities that appear appropriate to cater to groups of people [4]. During the quarantine, the recommendation is to avoid sharing, eating and sleeping in different spaces and to stay at least 4 metres away when passing or in shared toilet facilities if applicable [4].

Previous research in other settings showed that caregivers for COVID-19 patients expressed fears of getting infected, perceived their caregiving role as a burden, and compromised due to a lack of support or coping mechanisms to manage the role [6]. The experience of caring for a COVID-19 patient was deemed different from caring for patients with other conditions because COVID-19 was an emerging and unpredictable condition without a clear treatment

plan that exhibited fluctuating symptoms, unlike other conditions that have an outlined and fixed treatment regimen [7–9]. The care rendered was perceived as compromised because of a lack of information about the disease, a lack of health services support, and financial constraints [7, 8]. Caring for a COVID-19 patient resulted in unpleasant physical experiences from the use of disinfectants in the home [7], psychological stress over the illness [7–9], and in other instances, the psychological burden on caregivers [9], and negative social experiences from others [7, 8]. As coping mechanisms, caregivers depended on spirituality [7–9], their relationships [7], the maintenance of hope for a positive outcome [8], and adaptation to their role of caregiving [9].

The shift in the management of COVID-19 cases at home and the paucity of data on caregivers' experiences in managing patients within their homes necessitated the conduct of this study. This information will inform policymakers on how to draft leaflets and information bulletins to assist caregivers in managing their patients at home. This study explored the lived experiences of caregivers of COVID-19-infected individuals in Blantyre, Malawi.

## Methods

### Study design

This descriptive qualitative study was conducted from January to June 2021 in Blantyre, Malawi, among caregivers of confirmed SARS-CoV-2 cases that were enrolled in the SARS-CoV-2 infections, transmission dynamics, and household impact study in Malawi. Our study on caregiving experiences is part of a larger project, "SARS-CoV-2 infection, transmission dynamics and household Impact in Malawi (SCATHIM)", a cohort study that aimed at determining the transmission dynamics, determinants, and household socio-economic impact of SARS-CoV-2 infection in Malawian settings. The project had four work packages, namely: epidemiology, social science, health economics, virology, and immunology. Our study was in the social science and health economics package. We have reported the study following the COREQ Checklist (**S1 File**).

### Study setting

The study was conducted in both urban and rural Blantyre, in southern Malawi. As of 2018, Blantyre had a population of 1.25 million, with 66% of those residing in urban settings [10]. Blantyre is also a commercial city and has varied socio-economic statuses and residential areas. Most of the residents in these areas are formally employed or run formal businesses for their economic sustenance. Health services in Malawi are provided for free by the government, with a few provided by private and faith-based institutions. Most of the population resides in unplanned high-density urban and rural areas with limited water, electricity, and sanitation amenities, including houses in poorly demarcated areas close to each other. Most residents are informally employed or run small-scale businesses as a source of income. The minimum wage in Malawi is MK50,000.00 (41 USD). Since the start of the pandemic in Malawi, Blantyre has always been the epicentre of the COVID-19 pandemic. As of September 19 2022, Blantyre had registered 24,309 cases with a case fatality rate of 2.81.

As alluded to earlier, the study had an epidemiological work package that used the following demarcations: Blantyre rural, Blantyre urban low density, and Blantyre urban high density, and this study followed the same classification. This variation was important for the study because we hypothesised that different settings would accord caregivers varying experiences, which would broaden the nuances that caregivers would share in the study. The variation in settings was deemed a proxy for different economic statuses as well because it offered the

researchers an opportunity to recruit participants that had different socioeconomic rankings based on area of residence. Blantyre has one tertiary hospital and 36 Primary health facilities.

## Sampling and sample size

A purposive sample of 37 caregivers was drawn to achieve variation in the characteristics of participants. We considered variations in marital status, education level, location of residence, and gender. They were drawn from the enrolment register of the main study. All clients who had a COVID-19 positive result and did not require hospital admission were informed about the study, and those interested in taking part in the study were connected to the main study team. The epidemiology work package had an enrolment register where all patients with SARS-CoV-2 were captured, including their primary caregivers' information and their places of residence. The enrolment log was used to identify the caregivers who met the criteria for the qualitative study. We defined a caregiver as one who provides most of the patient's care. We recruited a caregiver 14 days after the patient was enrolled in the main study to give the caregiver adequate experience to draw lessons from. The study included caregivers above the age of 18 who were willing to participate, and primary caregivers of a COVID-19 patient enrolled in the main study, and we considered having participants with variations in characteristics. Thirty-seven caregivers were deemed adequate because Guest et.al. [11] argue that by the 12th interview, one would have captured 97% of the required information. We further observed similar themes in subsequent transcripts of the urban-based caregivers, with few transcripts for rural-based caregivers which was also in keeping with the pattern of the pandemic, with urban areas being worse affected than rural settings. We, therefore, increased the sample size for the rural-based caregivers to check if there would be different themes in the transcripts for the rural-based caregivers and to maximise the number of rural-based caregivers as they were few. We noticed no new information as we progressed with the additional interviews, suggesting we had reached data saturation [12].

## Data collection

We conducted 37 in-depth interviews among caregivers of SARS-CoV-2 cases from January to June 2021 using a semi-structured guide. Four seasoned qualitative research assistants, three female and one male, conducted the digitally recorded telephone interviews. At the time of the study, they were working as research assistants on the project. We opted for telephone interviews to minimise the spread of COVID-19. Researchers reviewed the enrolment log of the main study and purposely sampled from the list. The epidemiology work package information leaflet included a clause that primary caregivers may be contacted for participation in the social science work package later. Researchers would make an initial call to introduce themselves and the study, and if a potential participant was willing, the date and time for a telephonic interview would be set. The researchers identified themselves as research assistants in the SCATHIM study taking place at Kamuzu University of Health Sciences and explained that it was a study that recruited COVID-19-infected patients. Before the interview, the researcher obtained consent by reading out the consent form to the participant and recording the response from the participant as part of the audio data. After obtaining informed consent, the research assistant proceeded with the interview, following an interview guide (S2 File).

The interview guide was both in English and Chichewa (local language) to accommodate the participants' preferences (S1 File). The broad section of the guide inquires about their lived experiences and has subsections on challenges faced, solutions, and support received as they cared for their patient. Before the study, the guide was piloted among caregivers that we did not include in our analysis. The pilot aimed to ascertain the appropriateness and reliability of

the questions [13]. Data analysis began during data collection with the aim of determining the saturation and sufficiency of the size of the sample. Data saturation [12] was realised when there were no more new ideas from the participants when we got to the 34th interview. However, we conducted three more interviews to confirm saturation and maximise the number of caregivers from rural areas, who were initially few in the study. During the interviews, the researchers compiled field notes that included the process of the interview, the flow, and the participants' conduct during the interview [14]. There were no repeat interviews. Of the people that we approached, 13 refused participations for the following reasons: they found the study compensation not enough; they were not sure of the researchers' identities because of telephonic interviews; they were concerned with privacy and confidentiality issues because of telephonic interviews; and they denied that the patient had COVID-19. The average time of the interviews was 50 minutes, and participants were compensated for their time with MK2000 (1.95 USD).

## Theoretical framework: Theory of caregiving dynamics

The qualitative data on the lived experiences of the caregivers were analysed and categorised following a deductive approach and guided by the theory of caregiving dynamics, which is presented in Fig 1 [15]. This is a nursing theory and was developed by Loretta Williams. It has three constructs as follows: commitment, expectation management, and role negotiation. According to the theory of caregiving dynamics [15], commitment has four dimensions, which include enduring responsibility, prioritizing the patient, supportive presence, and self-care [15]. Commitment entails that a caregiver provides care to the patient irrespective of the

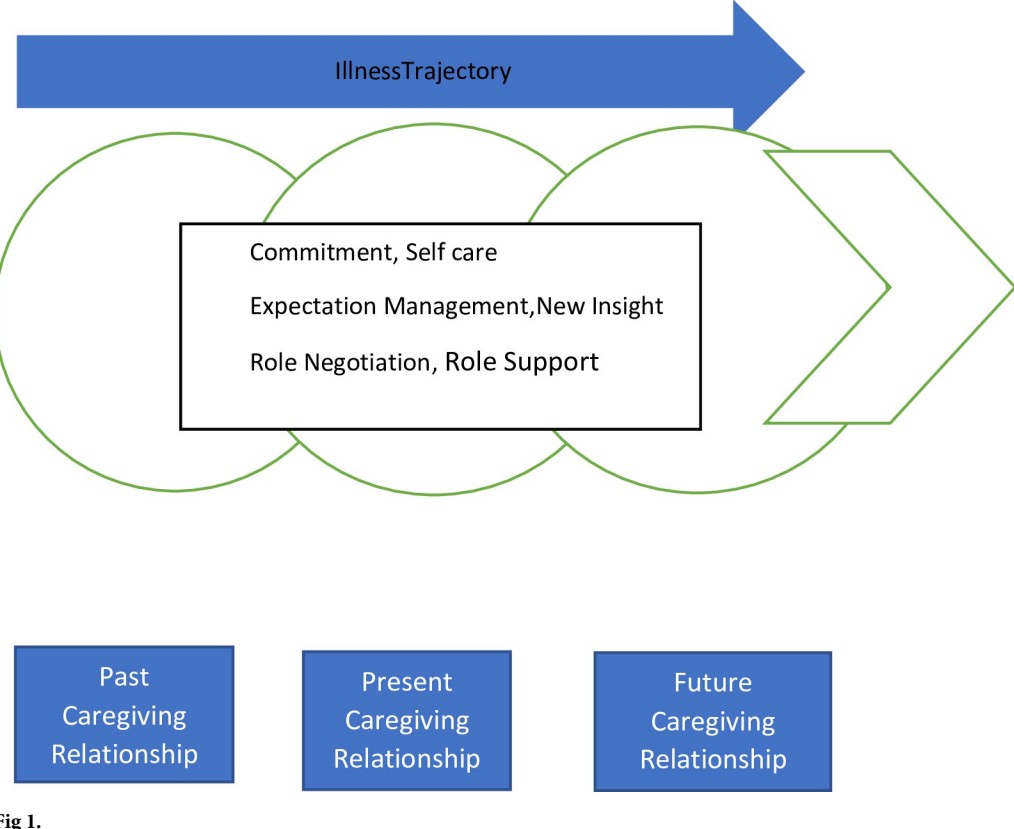

**Fig 1.**

risks, challenges, and lack of self-prioritisation that come with the role. Expectation management states that a caregiver and a patient have expectations, such as a worsening or a better prognosis of the patient's condition, and includes the roles that both the caregiver and the patient have to fulfill. Expectation management consists of envisioning tomorrow, getting back to normal, taking one day at a time, gauging behavior, and reconciling treatment twists and turns. Role negotiation entails a caregiver pushing and getting to grips with the caring responsibility, sharing responsibilities, attending to or listening to the patient's voice, and vigilantly bridging through negotiations with the health system in the care of the patient.

## Data management and analysis

All audio data were immediately transferred after the interview from the digital recorder, and stored in a password-protected Google Drive, and later transferred to a password-protected computer. We limited access and management to the researchers. All data were anonymised and participants were identified by a code and not their names.

The data were managed using NVivo 12 software (QSR International, Melbourne, Australia). All data that were in Chichewa were translated into English by a transcriber who is fluent in both languages. LSK conducted quality checks on randomly selected translated transcripts. Once data collection started, ALNM, MC, LSK, and DN reviewed three transcripts to develop the codebook. Each of them independently reviewed and coded transcripts and later held a meeting where they discussed the codes. Areas of difference were discussed to reach an agreement. The codebook was deductively developed following the construct of the theory of caregiving dynamics to initially code the transcripts in terms of the caregivers' experience, the challenges and measures that were taken to solve the challenges, and the actions they took in rendering care, such as being available, and sourcing support from others. Inductively, we coded the data based on what was found; for instance, economic aspects are not highlighted in the theory of caregiving dynamics but were more prominent in the data. Once the codebook was prepared, a social scientist who is familiar with NVivo independently and extensively coded all the data. The data coder trained the researchers on the codebook and had iterative meetings with them to consider adding additional codes or combining codes that were similar. MC and ALNM checked the coded data for accuracy and completeness. Then, the researchers examined the data set for patterns, taking into account both similarities and differences between the codes, and they organized all associated information into overarching themes.The main commitment, expectation management, and role negotiation elements from the theory of caregiving dynamics were used to name the themes [15].

Themes were produced from the data for both the data that fell inside the specified themes and the data that did not. For example, the data were used to inductively realize the subject of economics.

To make sure that the data backed up each of these themes, the audio and transcripts were compared. Themes with similar data were integrated, and all themes without supporting data were deleted.

## Ethics approval

Ethics approval was obtained from the Kamuzu University Research Ethics Committee, College of Medicine Research and Ethics Committee (CoMREC) (P.05/20/3046). The Director of Health and Social Services for Blantyre granted research approval to conduct the study in Blantyre. All participants provided verbal consent, cognisant that these were telephonic interviews. To limit physical contact and prevent the spread of COVID-19, a research assistant explained to a prospective volunteer that an agreement would be obtained over the telephone. A research

assistant also stated that the potential participant will be given the opportunity to have the IRB-approved consent form read out to them before deciding whether or not to participate in the study. This procedure was not observed because it was difficult to follow such procedures over the phone. The consent form was read to a participant in its entirety after the previously stated explanation, and he or she orally accepted to take part in the study. The audio recording shows the reading of it out loud and the decision-making process. The process of verbal consenting was approved by the Ethics Committee as a measure to prevent the spread of the COVID-19 infection.

## Results

### Characteristics of participants

The median age of the participants was 41, with an interquartile range of 32–51 years. Twenty of the caregivers were male and 27 of them were married. Twenty-seven had college-level education. Twenty-four caregivers were employed, and 15 resided in low-density urban areas of Blantyre (Table 1).

### Lived experiences of caregivers in the management of their COVID-19 patients

Participants stated various experiences in their caregiving roles. These were underpinned by the economic level of a family, and there were other interconnected dimensions such as commitment, expectation management, and the role of a caregiver. As such, the results of our study are underpinned by the economic status of a family, as illustrated in Fig 2. Fig 2 illustrates the interconnection of the factors with economic factors, as the platform of the

**Table 1. Characteristics of caregivers (N = 37).**

| Variable | Frequency |
|---|---|
| Age | Median 41; IQR 32–51 |
| Sex | |
| Females | 17 |
| Male | 20 |
| Marital Status | |
| Married | 27 |
| Widow | 1 |
| Single | 9 |
| Education | |
| College | 27 |
| Secondary | 8 |
| Primary | 1 |
| None | 1 |
| Occupation | |
| Employed | 24 |
| Business | 8 |
| Unemployed | 2 |
| Area of Residence | |
| Low density urban | 15 |
| High density urban | 12 |
| Rural | 10 |

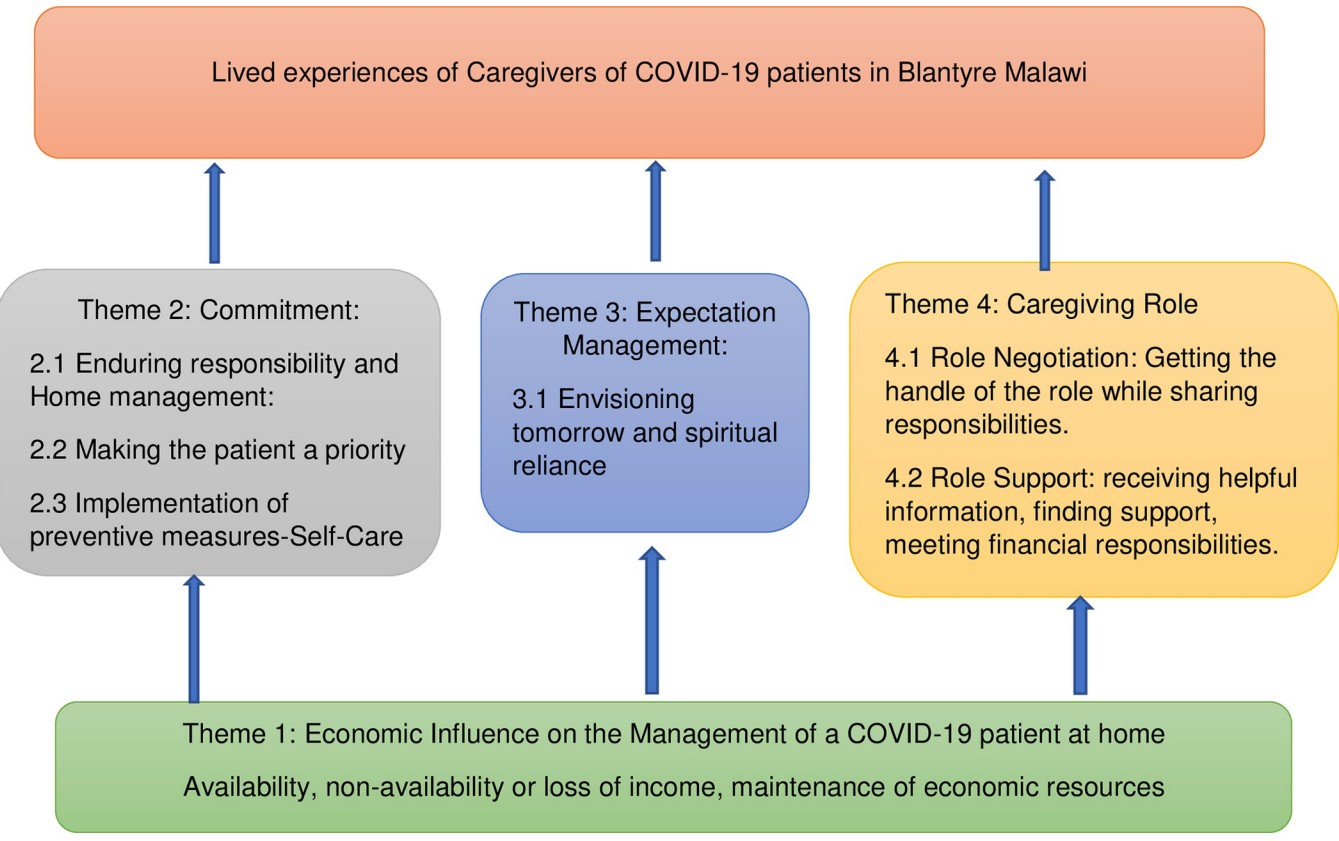

**Fig 2.**

caregiving experience. Fig 2 demonstrates that the economic factors within a family were a platform that determined the kind of lived experiences that caregivers had while nursing a patient at home. Commitment involved making their patient a priority for care in the context of infectious disease and the implementation of preventive measures as a form of self-care. Expectation management included envisioning what tomorrow would be like for the patient and caregiver. The role of the caregiver covered negotiating their role so that they are on top of their responsibilities and also getting support, which includes getting helpful information concerning the illness, getting others to help out as needed, and meeting financial responsibilities.

### Underpinning dimension: Economic influence on the management of a COVID-19 patient at home

In this study, the overall and commonly stated factor that largely influenced the lived experiences of caregivers was economic resources. Economic resources determined the adoption of preventive measures within households, such as the purchase of masks and sanitizers, access to relevant treatment, and the provision of adequate care to the patient. Some participants stated that they were economically challenged to care for COVID-19 patients at home, while others stated that they were able to offer the patient the necessary support. Participants narrated that their ability to live in a home where isolation was possible depended on their finances. Participants argued that the economic level of a family influenced the commitment level of a caregiver through their availability, which meant suspending their income-generating activities to provide care. Furthermore, it was stated that their finances influenced their expectations of the

future and their ability to recover financially when their patient got better. Economic levels also influenced the navigation of their role as caregivers and required them to depend on others to meet their financial demands if a family has such links; if not, then their role was negatively affected because they had low levels of income to execute their role effectively.

Participants that were not formally employed mentioned that economic challenges were more pronounced among families whose members were not formally employed but depended on small-scale businesses. They said that the needs of isolation as a precaution against COVID-19 worsened the financial struggles that caregivers and their patients experienced. Participants asserted that economic challenges were compounded in circumstances where a breadwinner was the index patient or when the primary caregiver was also the breadwinner and needed to isolate. They further narrated that aspects of their livelihoods that were affected included nutrition supplies and rental obligations, and in some instances, families went into debt to meet their financial demands.

> *"Our daily welfare wasn't good at all; we were affected so much because our only source of money was him (the patient), and we were living in hell, and it was difficult for us to find food. . ."* Participant 1041 Blantyre Rural Caregiver

> *"The major challenge was on the financial part; we were forced to improve our diet because we understood that one must have a good immune system to be able to fight the virus. So it is like compromising a lot of things, ranging from financial obligations to borrowing money from others."* Participant 1079 High-Density Urban Caregiver

On the other hand, financially stable caregivers, especially those that were formally employed and residents in the urban areas of Blantyre, stated that managing a COVID-19 patient at home enabled them to save because they never spent money on transportation.

> *"On my part, I saved on fuel because I am not that mobile, and I am not going to any drinking joint, . . . I cannot say that it impacted me negatively, but it did on my savings, so my savings were huge."* Participant 1007 High-Density Urban Caregiver

### Interconnected dimension: Commitment of the caregiver

The dimensions of commitment that informed the sub-themes include enduring responsibility, prioritizing the patient, supportive presence, and self-care [15]. There was an overlap between prioritizing the patient and offering a supportive presence and these have been combined in our results.

**Enduring responsibility.** The majority of the caregivers from all residential areas were committed to caring for their patients at home, despite some concerns about caring for patients with a 'highly contagious disease' at home instead of the hospital. Despite these concerns and a lack of information on how to care for the patients at home, they were committed to caring for them.

> *"At first, we thought that anybody who is positive for COVID-19 is supposed to be admitted to the hospital, so after telling us that we should be going home, we were very anxious to say: why take the patient back home and how will we be able to take care of her back home? Hence being anxious in the process. . ."* Participant 1012, Blantyre Rural Caregiver

> *"So, our only worry comes in because we do not know how we are supposed to be taking care of COVID-19 patients and wish the hospital staff were able to come and teach us how to do*

*it. . .I have never seen anywhere where the procedures for how people can take care of their patients are stated. . ." Participant 1023 Blantyre High-Density Urban Caregiver*

## Making the patient a priority and offering a supportive presence

As part of their commitment to care for their patients, most of the caregivers across all residential areas stated that the nature of the condition and the demands that ensued upon the family necessitated the prioritization of the patient's needs over everyone else in aspects concerning nutrition, medications, and comfortability. Caregivers prioritised the family's health over other financial demands like bills and rentals and ensured that they had finances to cater to their medical-related bills.

*"I think we just prioritised what was just necessary then, which was first buying the medication and then the food . . . On the businesses, I had to put them on a standstill" Participant 1009 Blantyre High-Density Urban Caregiver*

*"I just reduced some of the things that I was doing on other projects that I was working on and instead concentrated much more on our nutrition for us to remain healthy, like buying some fruits and vegetables . . . rushing to the hospital whenever we felt sick, even buying some fuel to take us to the hospital and sanitisers." Participant 1032 Blantyre Rural Caregiver*

A few caregivers who disclosed being infected with COVID-19 as well stated that despite being positive, they prioritised the care of the index patient over their own illness because of the severity of their patient's illness. Others stated that they continued taking the same medications their patient was taking and followed the same preventive measures such as steaming and using herbs as were prevalent at the time.

*"I was supposed to take care of them while I was also a patient and the rest of the people in the house were also waiting for my services." Participant 1026 Blantyre High-Density Urban Caregiver*

As an extension of prioritising the patient, caregivers were present and provided support to their patients. For caregivers to do this, they had to relinquish other responsibilities like business, work commitments, and social responsibilities, and others sent away their house help to prevent the spreading of the disease outside their home. This was common among all caregivers from all residential areas.

*"I am also privileged to be a treasurer of one of the fundraising committees at church, so what I did when my wife was found positive was meeting one of the members of the committee and tell them that "this is the case, my wife is positive, so I am handing over my things to you for, I think, the next 2 or 3 weeks. I will not be seen around, even at church or at home cells, during weekly meetings at church. I won't be around until I have or the family has been confirmed negative."Participant 1009 Blantyre High-Density Urban Caregiver*

The unfamiliarity and the rising COVID-19-related deaths that occurred at the time of the study influenced the presence of the caregiver and the level of prioritisation that was accorded to the patients. The news of COVID-19 was marred with fear among most patients and their caregivers, who were more concerned with the prognosis and outcome of the illness.

*"I was heartbroken and very disappointed with the news because of the way COVID-19 is killing people out there. So, I knew that if I was very disappointed, then I would make the patient very disappointed and worried too. . . I was very heartbroken when I first heard about the results."* Participant 1012, Blantyre Rural Caregiver

*"I was filled with fear since we hear rumors every day about people dying, but still, I tried to conceal my fear even though I was afraid too. I wondered what will happen next"* Participant 1015 Blantyre High-Density Urban Caregiver

## Self-care to prevent infection

Participants shared their experiences implementing preventive measures. Whereas others were able to implement measures like masking and isolation, there were challenges with implementation, especially among those in rural and high-density areas. Participants mentioned that in some cases, wearing a face mask was challenging because they feared that it would be construed as discriminatory and stigmatising. While some caregivers indicated that they wore face masks to prevent contracting the disease, other caregivers indicated that they did not wear a face mask because they perceived it as discriminatory, as shown below:

*"I think that it was just by the grace of God that we all came out negative after being tested. I could go into the bedroom without a mask and sometimes I could wear a mask, because when you put on a mask, it feels like you are stigmatising him or her."* Participant 1023 Blantyre High-Density Urban Caregiver

The use of sanitisers and masks also varied according to a family's economic status. Some caregivers in rural and high-density areas indicated that they used cloth masks and not medical masks because these were affordable and reusable, limiting the costs.

*"The main problem is the lack of resources that we can use in taking care of the patients because a lot of households do not have sanitisers, others are reusing disposable blue masks in the process, and most people are using cloth masks and I don't know if these masks, are recommendable."* Participant 1023 Blantyre High-Density Urban Caregiver

The maintenance of isolation measures within a household was also a challenge, especially for those in rural and high-density areas, particularly in homes that had limited space and large numbers of people. However, homes that had adequate space, like those in low-density areas, found it easier to isolate their patients and maintain those recommendations, as shown in the two quotes below:

*"We were anxious because the house is too small; it has two bedrooms, and already it contains six people. How are we going to go about it? How is she going to be isolated, looking at the present situation regarding the house?"* Participant 1012 Blantyre Rural Caregiver

*"[Patient] stayed in the sitting room because it is a bigger room than the other bedroom. When I cooked food, I would dish out my food and come to the sitting room, and then she would dish out hers and go to her room. On some days I would clean my room, and then she would clean, her room too."* Participant 1017 Blantyre Low-Density Urban Caregiver

In other cases, the patients were never isolated because of the inhumane connotations of observing these preventive measures. For some caregivers to provide a supportive presence,

they overlooked isolation measures because they were inconsistent with the way patients are managed, cognisant that ordinarily patients are never left on their own. Some caregivers stated that some amenities remained common for both the patient and caregivers, such as bathrooms, and others slept in the same room and bed. Over time, other caregivers got used to their role and were able to allow the patient to be isolated to prevent the spreading of the infection. One male caregiver narrated the experience of managing his sister; he explained the dilemma he had in applying isolation measures and how he finally got to do that after explaining to his sister the requirements:

> "I was very anxious about it (isolation) and worried because I was thinking about how she was going to be alone in the same house. I was thinking that it will be hard for me to do it, and it will be like I am stigmatising her in the process. How will I be able to go about it? How will I be able to provide for her? . . .." Participant 1012 Blantyre Rural Caregiver

Measures of isolation were extended to include the way family members related to one another emotionally. Families changed the way they related emotionally to one another as a means of minimising the transmission of the virus.

> "Of course, it can't be normal, and being worried wasn't going to work, but if you always have morning hugs, then there are no more morning hugs. I'm trying to maintain social distance even though it's not easy." Participant 1007 Blantyre Low-Density Urban Caregiver

The designation of specific cutlery and plates for a patient as a preventive measure was also a challenge in some homes, especially in cases where parents needed to have designated utilities for their children.

> "Everybody can use any cup at any time. So, it was difficult to control them. After all, it was our norm for anybody to use any cup or any plate within the house." Participant 1027 High-Density Urban Caregiver

## Interconnected dimension: Expectation management

This construct from the theory of caregiving dynamics [15] suggests that a caregiver envisions and longs for a return to normalcy. The tenets under this construct include envisioning tomorrow and spiritual reliance which are linked in the presentation of the results.

**Envisioning tomorrow.** Caregivers negotiated their expectations about the return to normalcy by encouraging their patients to take the necessary treatment for them to recovered soon. Caregivers applied various measures to ensure that their patients recover quickly. These measures included encouraging the uptake of nutritious meals at home, making plans by stocking up on what they needed if lockdown measures were enacted, and ensuring that the patient takes all the supplements like ginger and lemons.

> Okay, mostly, you know how you are trying to change your diet or the way you eat. Every member of the house was told, "You have to be drinking this (ginger and lemon drink)," which is not exactly sweet. . . they have to steam. . .we more or less had to have a lockdown, and we had to buy some other things that we had not planned to last for two weeks." Participant 1009 High-Density Urban Caregiver

> "Again, we could buy some of these drugs, which we didn't expect to buy all along. People were telling us to buy some Erythromycin, Zinc, and the like. In addition, we could rush to the

*hospital each time we observed that someone is not feeling well." Participant 1032 Blantyre Rural Caregiver*

Some of the caregivers' plans included financial strategies for returning to normal while mitigating the losses that were incurred during the pandemic. Participants envisioned that financial restoration would take some time, and feared that they may not fully recover from the losses they suffered.

*"I still have a couple of debts that I incurred because of this pandemic; I will be settling them gradually. That means that for us to go back to the way things were and for things to be alright, it will take time, and maybe some of these things will never be sorted out as they have already veered off." Participant 1042 Blantyre Rural Caregiver*

Although caregivers maintained a positive attitude and expected positive outcomes, they stated that they were affected by the perceptions of their neighbours who expressed a gloomy picture of the possible outcome of the sickness, such as death. This was aggravated by the increase in the number of deaths secondary to COVID-19.

*"Another problem we faced was the rumors that people were spreading about us, saying, "We do not know if they are going to make it tomorrow, or have you heard about them? They have got COVID-19 and so on." Participant 1019 Blantyre High-Density Urban Caregiver*

Some caregivers depended on their faith as a coping mechanism, especially in instances where family members remained COVID-19-negative while nursing a positive patient in the home.

*"I think that it was just by the grace of God that we all came out negative after being tested." Participant 1023 Blantyre High-Density Urban Caregiver*

*"As a person who once had COVID-19 disease, when I heard that people are dying as a result of COVID-19, then I started to think that maybe I will also die of COVID-19 in the process, but by the Grace of God, I am still alive today and I don't have COVID-19 in me." Participant 1034 Blantyre Low-Density Urban Caregiver*

## Interconnected dimension: Caregiving role

Participants illustrated that their caregiving role encompassed their understanding of what is expected of the role, which is role negotiation. The other aspect of their caregiving role included the support they received as caregivers. Thus, the subthemes under this theme include role negotiation and role support [15].

**Role negotiation.**   Caregivers expressed that they navigated their role by getting a handle on the expected responsibilities as far as the provision of care was concerned, sharing responsibilities to ease up their role, and receiving support to perform their role. Even though observing isolation was a challenge in large families, reliance on household members and social networks helped ease the caregiving roles of COVID-19 patients. Caregivers stated that while managing their patients at home, they shared responsibilities within the home and among other family members outside the home. These responsibilities included purchasing items that were needed while the patient and the family were in isolation. Other domestic responsibilities were shared within the home.

*Interviewer*: *So how were you managing your daily welfare, for example, your daily household chores*?

*Respondent*: *It was done perfectly because we were able to sweep the house, somebody would wash the plates, and everything was done accordingly.*

*Interviewer*: *Okay. Even during her sickness*?

*Respondent*: *Yes, and to be honest with you, she wasn't that serious because I didn't see her down. She would complain of some body pains, but she wasn't down.- Participant 1018 Blantyre High-Density Urban Caregiver*

On the other hand, caregivers in small families stated that they had to bear domestic responsibilities, especially in cases where a female partner was a patient. While men normally invite female relations to look after their female partners when they are ill, the requirement to prevent transmission to others outside the home challenged the men to embrace domestic roles. As such, male partners had to assume traditionally designated female roles in the home. Some men felt embarrassed to undertake domestic responsibilities designated for women outside the home, where neighbors could see them. They would therefore undertake such responsibilities in the early morning or late at night to avoid being seen by neighbors.

*"[Home management] is quite challenging because, being a man, I am not used to buying these things (household stuff) or preparing food for the day. Yeah, it was quite a challenge. . .but I was doing that at odd hours because I didn't want people to see me or notices that I had gotten out." Participant 1027 Blantyre High-Density Urban Caregiver*

*"We were only the two of us, and I had to work up every morning, doing everything. . .I had to wake up early in the morning to make breakfast and even clean the house and wash the utensils." Participant 1003 Blantyre Rural Caregiver*

## Role support

Caregivers sought various kinds of support to enhance their role as they cared for a patient. The support they sought was in the form of finding helpful information, finding others to support other activities, and meeting financial responsibilities. These responsibilities included purchasing items that were needed while a caregiver and the patient were in isolation.

**Receiving helpful information.** Some caregivers were concerned that they had limited information on caring for their patients and had hoped that healthcare workers would teach them about the management of COVID-19 patients at home.

*"When a patient has been found positive for COVID-19, they (health workers) should inform the family of the medicine to buy." Participant 1030 Blantyre High-Density Urban Caregiver*

In the absence of concrete information platforms, caregivers sought information from their social networks. Participants stated that the lack of readily available information due to the novelty of the condition forced them to solicit information from those within their social circles, which included healthcare workers and other members of the community.

*I*: *How did you overcome those fears that you have had at that time*?

*R*: *I met some people and some nurses who used to encourage me in my spiritual life; they were saying that I wasn't supposed to do things like those*: "*Just accept that you have COVID-*

*19 with all your heart, you will not die, and you will just be fine. Participant 1034 Blantyre Low-Density Urban Caregiver*

Other caregivers attested to being counselled at the point of testing and being informed about the nature of the illness and the benefits of minimising stress while managing the condition. This counselling assisted some caregivers in communicating positive results to their patients.

*"However, since they gave us counselling during the COVID-19 testing, they told us that the illness doesn't need one to be stressed out;there's no need for anxiety. . . So, since they already told us about it, I composed myself because I was the one to inform the patient about the outcome of the COVID-19 testing." Participant 1002 Blantyre Low-Density Urban Caregiver*

**Finding support to manage household needs and businesses.**   Caregivers depended on others to get essentials for them, including caring for the patient to allow a caregiver to attend to their work. Other caregivers stated that school holidays eased up their roles because they only focused on their patients and were relieved of the demands associated with school. In other cases, the housekeeper assisted in managing the home while observing preventive measures so that they were not in contact with the patient but with the caregiver only.

*"I was managing it (caregiving role) because at first I was the one taking care of her for a few days, then a few of her relatives came to help in taking care of her, like when I went to work, then her relatives were the ones taking care of her. Participant 1030 Blantyre High-Density Urban Caregiver*

*"The problems were there since we needed to go and deliver the rice so that people could take it. The fritters that we sell outside require us to go and buy flour; those errands are on hold. The process of making fritters requires one to buy resources like firewood, oil, and the like, and if you don't have them, where can you go? You can't 'go to another person since they are already discriminating against you?" Participant 1033 Blantyre Low-Density Urban Caregiver*

Although some were able to find a person and had the finances to get essentials, some found that difficult as they had no one they could depend upon, and if available, they found themselves dependent on the person's availability to assist them. Reliance on others was a problem when the person providing support also had a COVID-19 patient or had suffered a loss from the same.

*"The only challenge was the issue of asking someone to go and buy something for you at the shops; we relied on that person's availability to do that without forcing him or her." Participant 1002 Blantyre Low-Density Urban Caregiver*

*"It was a huge burden on my sister because her husband died of COVID-19 and then her two sisters had COVID-19 as well, so I was feeling sorry for her because we all depended on her to give us support. . ." Participant 1029 Blantyre High-Density Urban Caregiver*

The lack of medicines and support in the public facilities forced caregivers to spend money and access the medicines to treat the symptoms they presented with in private facilities.

"*I was using my means, like using my money to go to the hospital, since they offer the testing without providing the treatment. So, we could just go to any private clinic and get the cough and flu drugs.*" Participant 1032 Blantyre Rural Caregiver

**Meeting financial responsibilities.** Other caregivers narrated that they had no one to support them, and they managed the financial responsibilities by themselves. This was easier for families that were formally employed, unlike those that were not employed or were running small-scale businesses.

"*We were just waiting for the salary at the end of the month, and there was nobody who came and assisted us during that time. . .it was just me and my children and my husband, but I had a lot of problems. You can imagine that it was only him (husband and patient) who is a breadwinner, and then he got sick, yet we only rely on the same person who was sick, and nobody came to assist us.*" Participant 1092 Blantyre Rural Caregiver

Another caregiver narrated their experience managing with minimal resources in this way:

*Respondent*: We were just staying like that, and we would just prepare

porridge with salt only, and we would eat it, even eating nsima with vegetables only.

*Interviewer*: Did you reach that level?

*Respondent*: Yes, do you expect our relatives to be helping us every day? We reached that far, and there was nothing we could do about it. . ., you can't force a relative to buy something for you when he or she doesn't have anything to offer. . . we would eat whatever is available. Participant 1019 Blantyre High-Density Urban Caregiver

Other caregivers let their friends run their businesses and would only collect the money as needed. As the patients got better, some caregivers resumed their income-generating activities so that they could rebuild their economic base.

"*I left my business with some friends so that they could be selling them on my behalf, and I was able to get some money after the sales. Participant 1012 Blantyre Rural Caregiver*"

## Discussion

The main finding from this study on the lived experiences of caregivers of COVID-19 patients was that a family's economic status largely influenced their caregiving roles and abilities. Furthermore, the results show that caregivers were committed to their role despite being ill-prepared and scared about managing a COVID-19 patient at home. In their commitment to care for the patient, they prioritised the health of their patient by ensuring that they were present and meeting the nutritional and medical demands related to the illness. In some instances, they found practicing preventive measures challenging because of financial challenges and cultural factors. The expectation of having their patients healed was shaken by the increasing number of deaths that occurred in their areas of residence, and they depended more on their spiritual links to avoid despair. Caregivers managed their role by sharing responsibilities, getting better at their role with time, and getting support from social networks to cover aspects like accessing helpful information and finding support to help meet other needs and financial responsibilities.

The finding that the caregiving experience was influenced by economic position, as demonstrated in this study, is consistent with previous results that a lack of resources led to unmet needs for the family [7, 16, 17], particularly in cases where the patient served as the family's primary provider and was not formally employed [7]. Other caregivers in Iran also encountered the excessive spending by families that was seen in our study; these caregivers spent more money on nutrition and medical needs [8]. Our results differ slightly from earlier findings, where there was a significant request for the government to support the families [7]. Despite wanting such support, the participants in our study expressed uncertainty about its viability. We assert that with careful preparation and assessment of the ultra-poor and vulnerable households, the Malawi government could implement social support for such families. There is a need to adopt a vulnerability assessment scale that can assess a household's access to cash and food and that can apply to a Malawian setting in order to correctly identify ultra-poor and vulnerable households [18]. Cash transfers are a form of social protection, and an evaluation of their effects found that they have positive effects, such as an improvement in health and nutrition indicators and a greater utilization of healthcare facilities [19]. Unconditional cash transfers, in particular for Malawi, have demonstrated effectiveness in promoting improvements in nutritional status and health-seeking while ill [20], which would be congruent with the goals if applied in the setting of COVID-19 infection. Industrialized nations have supported households affected by COVID-19 with flexible leave days, financial support, and protective wear. Although the Malawi COVID-19 Urban Cash Intervention was implemented, none of our study participants had access to it because it took place between April and June of 2020, prior to the start of the study. Our findings offer evidence that can be used to design future support for people affected by pandemics [21].

Our study showed that, despite their fears, caregivers were dedicated to taking care of their patients. This result is in line with that of an Iranian study, which found that caregivers were terrified of the unpredictable nature of the diseases and the fluctuating symptoms that their patients presented with [7]. Additionally, previous research revealed caregivers' fear of getting the virus [8], which was also evident in our study. Although caregivers voiced these concerns, fiscal constraints and the concern over coming off as prejudiced toward the patient prevented the implementation of preventive measures. Our study took place in Malawi during the second wave of COVID-19, when more people were dying and getting sick, which raised caregivers' anxiety levels. To prevent alarming those who are sick or caring for such patients, the practice of making the fatalities public as a way of informing the public on the trends of the pandemic should be done carefully [22]. The challenging social relationships, as expressed in this study, build on findings from earlier studies that demonstrated that COVID-19 disrupted relationships due to the implementation of isolation measures that imposed a lack of interaction and impeded the support that a family could have received [23]. Additionally, the installation of isolation measures stressed out caregivers, as was previously observed [24, 25], and it also had an impact on their way of life, particularly their ability to get necessities or run businesses, as was previously described [25]. An earlier review reported that people who were isolated and caring for a COVID-19 patient had the most anxiety about the illness and its implications [26]. Our study showed that it was challenging to enforce isolation measures in households with limited space, which is consistent with claims made before that confined spaces render it difficult to implement preventative measures [27]. In order to effectively point caregivers in the right direction, we contend that the existing Malawi COVID-19 home management guidelines should differentiate isolation methods across the different types of households [28]. Supporting vulnerable families with protective clothing, such as masks with soap and water or sanitizers, is necessary to achieve better rates of utilization of preventative measures while delivering care at home [16].

This study's findings about the absence of resources, knowledge, and guidelines for managing COVID-19 patients are in accordance with those of earlier research [8]. The lack of assistance from healthcare professionals is still a source of concern, as this study has shown since it is likely to lead to stress [29]. According to earlier research, caregivers are likely to follow instructions for managing their patients [16] and urge the provision of professional support when they provide the service [30]. Going forward, it is imperative to disseminate information on COVID-19 extensively and make it available through a variety of channels, such as fact sheets and follow-up phone calls to caregivers [31]. In contrast to social media, which is overflowing with false information [31], this information should be actionable, hope-oriented, and presented in a visual style [32]. It should also originate from reputable sources, such as health institutions [22]. Previous research has indicated that some caregivers acquire contradictory information, which may compromise the care they provide and increase their anxiety levels [9]. In the context of COVID-19, providing the appropriate information is crucial because it reduces the risk of stigma and confusion, which are triggered by incomplete and inconsistent information [22].

The spiritual relationships that caregivers in our study drew strength from have been previously reported [8]. Going forward, caregivers in pandemics like COVID-19 in Blantyre, Malawi, can benefit from the provision of psychological and spiritual support by the religious fraternity and other public institutions [22] that is preceded by the appropriate training. Religious institutions are reputable organizations with the ability to support family caregivers at home and boost patients' and caregivers' levels of hope as a coping mechanism [22, 33]. To reduce stress and promote psychological health, social support is a crucial component of care that should be included in the management of COVID-19 cases [34].

## Strengths and limitations

The use of telephone interviews limited the observations that could be made by the caregiver as they narrated their experience; however, the research assistants mitigated this limitation by probing for more information. The strength of this study lies in collecting information from caregivers from varied residential areas, which broadened the scope of responses. Our sample had more participants who had college-level education, which could make the results more applicable to such groups, and future studies should purposefully include those with limited education levels. Although we deliberately recruited more participants to ensure that we had an equal representation of those who dwelled in rural areas, which was a proxy of economic status in this study, we remained with a lower number of participants from rural settings. Future studies should tease out the baseline economic activities in detail so that they challenge or support the assertions in this study, and they could consider limiting the study to rural areas alone and conducting a case study. The refusal rate in the study was high, and this could be because participants were skeptical about privacy and confidentiality in telephonic interviews. On the other hand, telephone interviews could not allow us to assess the circumstances in which one lived or enable potential participants to verify the authenticity of the researchers. Future studies should focus on specific reasons for enrolling or declining enrolment in telephone interviews in emergencies.

## Conclusion

The economic status of a household determined the experiences of caregivers as they managed their COVID-19 patients at home. Caring for confirmed cases of COVID-19 demanded commitment from the caregivers while ensuring that the transmission of the virus was minimised. There is a need to support households in isolation with the right information on how to

manage their patients and streamline social support for the ultra-poor. The existing community structures are a platform that can be used to support the affected households.

## Supporting information

**S1 File. COREQ checklist.**
(PDF)

**S2 File. Interview guide.**
(DOCX)

## Acknowledgments

This manuscript is part of the "SARS-CoV-2 infection, transmission dynamics and household Impact in Malawi (SCATHIM)". The authors are grateful to the study participants for their voluntary participation, and the Director of Health and Social Services for Blantyre for institutional support.

## Author Contributions

**Conceptualization:** Alinane Linda Nyondo-Mipando, Victor Mwapasa.

**Data curation:** Leticia Suwedi-Kapesa, Marumbo Chirwa.

**Formal analysis:** Alinane Linda Nyondo-Mipando, Deborah Nyirenda, Leticia Suwedi-Kapesa, Marumbo Chirwa.

**Funding acquisition:** Victor Mwapasa.

**Investigation:** Alinane Linda Nyondo-Mipando, Deborah Nyirenda, Leticia Suwedi-Kapesa, Marumbo Chirwa, Victor Mwapasa.

**Methodology:** Alinane Linda Nyondo-Mipando, Deborah Nyirenda.

**Resources:** Alinane Linda Nyondo-Mipando.

**Supervision:** Victor Mwapasa.

**Validation:** Alinane Linda Nyondo-Mipando.

**Writing – original draft:** Alinane Linda Nyondo-Mipando.

**Writing – review & editing:** Alinane Linda Nyondo-Mipando, Deborah Nyirenda, Leticia Suwedi-Kapesa, Marumbo Chirwa, Victor Mwapasa.

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
