## [Decision Letter · Decision Letter 0]

21 Feb 2023

PGPH-D-22-02004

“Why take the patient back home?”:exploring the lived experiences of caregivers of COVID-19 infected individuals in Blantyre, Malawi.

Dear Dr. Nyondo-Mipando,

Thank you for submitting your manuscript to PLOS Global Public Health. After careful consideration, we feel that it has merit but does not fully meet PLOS Global Public Health’s publication criteria as it currently stands. Therefore, we invite you to submit a revised version of the manuscript that addresses the points raised during the review process.

The manuscript has been evaluated by two reviewers, and their comments are available below.

The reviewers have raised a number of concerns that need attention. They request additional information on methodological aspects of the study (e.g. the justification of sample size and cessation of data collection),and request additional information such as information on the Malawian health care and home care guidelines for those with covid-19. 

Could you please revise the manuscript to carefully address all the concerns raised?

We look forward to receiving your revised manuscript.

Kind regards,

Katrien Janin

Staff Editor

Journal Requirements:

1. In the ethics statement in the Methods, you have specified that verbal consent was obtained. Please provide additional details regarding how this consent was documented and witnessed, and state whether this was approved by the IRB.

3. Please provide separate figure files in .tif or .eps format only and remove any figures embedded in your manuscript file. Please also ensure that all files are under our size limit of 10MB.

4. Your manuscript is missing the following sections: Abstract. Please ensure these are present, and in the correct order, and that any references to subheadings in your main text are correct. An outline of the required sections can be consulted in our submission guidelines here:

https://journals.plos.org/globalpublichealth/s/submission-guidelines#loc-parts-of-a-submission

5. We have noticed that you have uploaded Supporting Information files, but you have not included a list of legends. Please add a full list of legends for your Supporting Information files after the references list. 

6. In the online submission form, you indicated that "The datasets used and/or analysed during the current study are available from the corresponding author on reasonable request". All PLOS journals now require all data underlying the findings described in their manuscript to be freely available to other researchers, either 1. In a public repository, 2. Within the manuscript itself, or 3. Uploaded as supplementary information.

Additional Editor Comments (if provided):

Reviewers' comments:

Reviewer's Responses to Questions

**Comments to the Author**

1. Does this manuscript meet PLOS Global Public Health’s publication criteria? Is the manuscript technically sound, and do the data support the conclusions? The manuscript must describe methodologically and ethically rigorous research with conclusions that are appropriately drawn based on the data presented.

Reviewer #1: Yes

Reviewer #2: Partly

2. Has the statistical analysis been performed appropriately and rigorously?

Reviewer #1: N/A

Reviewer #2: N/A

3. Have the authors made all data underlying the findings in their manuscript fully available (please refer to the Data Availability Statement at the start of the manuscript PDF file)?

Reviewer #1: No

Reviewer #2: No

4. Is the manuscript presented in an intelligible fashion and written in standard English?

Reviewer #1: Yes

Reviewer #2: Yes

5. Review Comments to the Author

Reviewer #1: In this manuscript, the authors describe the experiences of covid-19 caregivers in and around an urban centre in Malawi. The authors have fairly described the previous literature on the topic, contextualizing their results. Thank you to the authors for this thoughtful manuscript which describes important experiences for which there has been little research.

Major Revisions Suggested

1. Overall the methods are well described, but there are 2 areas which need strengthening. Firstly, the analysis method was described as thematic, but appeared to be a deductive analysis based on the theory of caregiver dynamics. The "themes" seemed to be entirely consistent with the concepts in the theory. This is also reflected in Figure 1 which does not seems to contain little study-specific information, but mostly just represent the theory. I suggest: 1) introducing the theory at an earlier point in the methods section and giving it its own heading, 2) calling this a deductive analysis, 2) having 2 figures in the paper, one which represents the theory (with no data) and the other figure which displays how the data from study supports the theory.

2. There is very little information in the manuscript on the Malawian health care and home care guidelines for those with covid-19. It would have been very useful to know if there were any guidelines on what kinds of patients were admitted to hospital (versus sent home with caregivers), the quarantine guidelines (if any) for household contacts of covid-19 patients, government recommendations (if any) for in-home distancing, cleaning, masking, etc. A fuller description of these items would strengthen the manuscript.

3. A key underlying theme of the study is that the family's economic resources influenced all aspects of caregiving. This was displayed as a central/fundamental concept in Figure 1, but explicitly acknowledged as a key concept in the written results. Adding a sentence describing how this concept under-girded the others would be more clear. Given the centrality of this concept, there needs to be more information in the methods and results regarding the economic status of the participants.

The authors frequently mention the impact of caregiving on poor and ultra-poor households, but we do not know how many caregivers from these households participated in the study (neighborhood was used as a proxy, but is not particularly descriptive). How were covid-19 cases recruited to the main study? Could this have influenced the SES of the caregivers? What kind of variation was Participants must have had a access to a phone to participate, meaning that they may not have been "ultra-poor". Also 27 participants were college educated, which may indicate higher economic status. Also there is a suggestion that participants were compensated for their interview, but there is no mention of the amount, which may have influenced participation. These methodological consideration and thier impact on the results should be further discussed as a limitation to the study (if in fact it was a limitation).

Minor Revisions Suggested

1. Justification of sample size and cessation of data collection requires more support. Were concepts fully developed? Why did the AUTHORS (who are familiar with the setting and the topic being studied) expect that the sample size would be adequate?

2. The interview guide was not described and the supplemental file 2 was not present. The text should be able to stand alone and should contain a description of the interview guide and how it was developed? Was the theory of caregiver dynamics also used to develop the interview guide?

3. Criteria used for purposive sampling should be described

4. More information on the process of transcription and translation is required. Was any verification of transcription and translation completed? Was the transcript verbatim?

5. More information was needed on the "coder". Coding is a key component of qualitative research and the training of this person and how the research team worked with this person was not sufficiently described.

6. Another 1- 2 sentences on how the researchers introduced themselves to the participants and the study prior to informed consent would be helpful.

7. The heading scheme is the results section is confusing. I would encourage the authors to explicitly identify the number of major and minor themes and then revise the heading scheme to make this more clear.

8. In the results section, I suggest using distinct terms to differentiate the practice of caregivers remaining in the home (to prevent transmission of covid-19 to non household members) and family members keeping to separate parts of the home (to prevent transmission of covid-19 to other household members). I believe the authors used the word isolation for both and this was somewhat confusing

9. In the results, several quotes appear disconnected from the sentence immediately preceding the quote. For example on the bottom of page 19 (older children assume household tasks) and the middle of page 21 (people in social networks gave advice on how to care for covid-19 patients). I suggest to revise the results so there is a very tight connection between the idea expressed just before the quote and the contents of the quote. This will strengthen the credibility of your results (and conclusions). I also suggest that you include a quote which more clearly expresses the idea that the caregiver's future expectations were challenged by the high case fatality rate associated with covid-19. The data supporting this claim (which was mentioned in the abstract and the discussion) is not strong.

10. In the abstract, the second sentence about WHO guidelines does not add to the abstract and could be deleted.

Reviewer #2: Topic: Why take the patient back home?”:exploring the lived experiences of caregivers of COVID-19 infected individuals in Blantyre, Malawi.

This is an important paper for Malawi given the guidelines on managing non severe cases of COVID 19 at home. It can support improvements in this area to help caregivers cope better with more standard information and support.

Abstract:

It is well written with a clear summary of the study.

Introduction:

Capitalise the I in the introduction section heading.

Generally, well written with studies in other settings reviewed to provide context to this paper.

Methods

1st paragraph. There two statements that are repeated. “Most residents are informally employed or run small-scale, informal businesses as a source of income…” You need to delete one. Either the 1st or the one coming sentences later.

 

Four seasoned qualitative research assistants, three female and one male namely MC, WN, LC, and RS conducted the digitally recorded telephonic

interviews. - Are these co-authors? Some initials are not anywhere in the author list.

How much was the compensation? This is not provided, and yet one of the reasons for declining to participate was a low compensation amount. Please state clearly how much it was. May be under ethical considerations?

13 refusals out of this sample size approached is a substantial number. It is a very high refusal rate. It is also possible that those who declined because of compensation not being enough are of a particular social economic status. Could this introduce any bias to the study? Selection bias may be? Could this be acknowledged? It could also be because these were phone interviews in a setting where people value face to face interviews. No doubt, one reason was being uncertain of interviewer’s identity

Page 9, 1st sentence. Twenty-four caregivers were employed and 15 resided in low density urban areas of Blantyre, respectively. - What does respectively mean in this sentence?

Page 10, last sentence. “Figure 1 below illustrates the interconnection of the factors with economic factors as the platform of the caregiving experience. - Take out the “below” because you do not determine where the figure will be located in the final paper. A figure reference is all you need I think. The figure may be location 2 pages away from this sentence for example, in the final paper.

Results

This comment may mean a modest revisit of the results. Although I am not proposing quantifying qualitative results, it helps to put some magnitude in the findings. How many participants mentioned somethings may be important to gauge how important that result is. It is not clear to know from the results currently if everything mentioned was coming from majority of participants, minority or even deviant cases or just one. Also providing for the categorization in the findings as it is in the sample would be important. Are some issues more pertinent for rural participants, or urban for example? This I believe is vital. Just providing two quotes, one from urban and another rural does not make the submission necessarily relevant to both. It could be just that one participant quoted from urban out of 15 that said something.

Page 12: On the other hand, financially stable caregivers especially those that were formally employed stated that managing a COVID-19 patient at home, enabled them to save because they never spent money on transportation.

“On my part, I saved on fuel because I am not that mobile, I am not going

to any drinking joint, … I cannot say that it impacted me negatively, but it

did on my savings, so my savings were huge.” Participant 1007 High-

Density Urban Caregiver

This quotation does not seem to bring out the explanation that is synthesized. The participant says they are not that mobile anyway. So as a a caregiver, if they were not mobile from prior, how is this changing when they have someone to care for at home? I don’t see how the quote relates directly to enabling the participants save. Is there a better quotation, since this is a theme anyway? I believe you have several.

Page 13: Caregivers prioritised the family's health over other financial demands like bills and rentals and ensured that they had finances to cater to their medical-related bills. - What is a Bill? You mention bills here twice. What did you mean by the 1st bills stated, because medical costs are also bills.

Page 14: Some caregivers said that despite being positive for COVID-19 as well, they prioritised the care of the index patient over their illness because of the severity of the illness of their patient. - what do the authors mean by the severity of the illness here? It is clear that all severe cases are managed at health facilities, as stated in the introduction. Now they introduce two points here. The index patient and severity. You need to explain what was meant clearly. If the index patient was not as severe as the other patients, what would happen? If the patient was severe, were they managed at home?

There are several areas where the authors just provide a one line result and then quotation(s). They need to synthesise the results and provide some detail from the data.

For example on page 17: Families changed the way they related emotionally to one another as a means of minimising the transmission of the virus. - It is not clear how they were emotionally relating before, and how the caregiving changed that relationship. Only providing a quote from one participant does not explain this.

Page 19: Some caregivers depended on their faith as a coping mechanism, especially in instances where the family members remained COVID-19 negative while nursing a positive patient in the home. – I don’t see how this is a coping mechanism. The quotes provided only show what happened after. They do not indicate that during the care process they relied on their faith. May be it is the writing that needs modification.

Some men felt embarrassed to undertake domestic responsibilities designated for women outside the home where neighbors could see them. They would therefore undertake such responsibilities at odd hours to avoid being seen by neighbors. - What are those odd hours? Provide more context from the data. I also do not think the quotation that follows would have “odd hours” mentioned in the local language. Is that the case? Or were there specific times mentioned.

I notice inconsistences in quotes that have Interview and respondent included. In one cases you provide these in full, and in another you provide initials. Pages 20, 21 and 24

The second quotation under Finding support to manage household needs and businesses does not relate to the synthesized text. On page 22

Under strengths and limitations its mentioned that: The strength of this study lies in collecting information from caregivers from varied residential areas which broadened the scope of responses.. – But, the study findings do not provide that distinction. This would cease to be a strength if the results do not provide this distinction. They are all bundled up currently.

6. PLOS authors have the option to publish the peer review history of their article (what does this mean?). If published, this will include your full peer review and any attached files.

**Do you want your identity to be public for this peer review?** For information about this choice, including consent withdrawal, please see our Privacy Policy.

Reviewer #1: No

Reviewer #2: **Yes: **Simon Peter Sebina Kibira

---

## [Decision Letter · Decision Letter 1]

1 Jun 2023

PGPH-D-22-02004R1

“Why take the patient back home?”: exploring the lived experiences of caregivers of COVID-19 infected individuals in Blantyre, Malawi.

Dear Dr. Nyondo-Mipando,

Thank you for submitting your manuscript to PLOS Global Public Health. After careful consideration, we feel that it has merit but does not fully meet PLOS Global Public Health’s publication criteria as it currently stands. Therefore, we invite you to submit a revised version of the manuscript that addresses the points raised during the review process.

We look forward to receiving your revised manuscript.

Kind regards,

Miquel Vall-llosera Camps

Staff Editor

Journal Requirements:

1. In the ethics statement in the Methods, you have specified that verbal consent was obtained. Please provide additional details regarding how this consent was documented and witnessed, and state whether this was approved by the IRB.

Reviewers' comments:

Reviewer's Responses to Questions

**Comments to the Author**

1. If the authors have adequately addressed your comments raised in a previous round of review and you feel that this manuscript is now acceptable for publication, you may indicate that here to bypass the “Comments to the Author” section, enter your conflict of interest statement in the “Confidential to Editor” section, and submit your "Accept" recommendation.

Reviewer #1: (No Response)

Reviewer #2: All comments have been addressed

2. Does this manuscript meet PLOS Global Public Health’s publication criteria? Is the manuscript technically sound, and do the data support the conclusions? The manuscript must describe methodologically and ethically rigorous research with conclusions that are appropriately drawn based on the data presented.

Reviewer #1: Partly

Reviewer #2: Yes

3. Has the statistical analysis been performed appropriately and rigorously?

Reviewer #1: N/A

Reviewer #2: N/A

4. Have the authors made all data underlying the findings in their manuscript fully available (please refer to the Data Availability Statement at the start of the manuscript PDF file)?

Reviewer #1: Yes

Reviewer #2: Yes

5. Is the manuscript presented in an intelligible fashion and written in standard English?

Reviewer #1: Yes

Reviewer #2: Yes

6. Review Comments to the Author

Reviewer #1: (No Response)

Reviewer #2: (No Response)

7. PLOS authors have the option to publish the peer review history of their article (what does this mean?). If published, this will include your full peer review and any attached files.

**Do you want your identity to be public for this peer review?** For information about this choice, including consent withdrawal, please see our Privacy Policy.

Reviewer #1: No

Reviewer #2: **Yes: **Kibira Sebina Simon Peter

---

## [Decision Letter · Decision Letter 2]

1 Aug 2023

PGPH-D-22-02004R2

“Why take the patient back home?”: exploring the lived experiences of caregivers of COVID-19 infected individuals in Blantyre, Malawi.

Dear Dr. Nyondo-Mipando,

Thank you for submitting your manuscript to PLOS Global Public Health. After careful consideration, we feel that it has merit but does not fully meet PLOS Global Public Health’s publication criteria as it currently stands. Therefore, we invite you to submit a revised version of the manuscript that addresses the points raised during the review process.

We look forward to receiving your revised manuscript.

Kind regards,

Miquel Vall-llosera Camps

Staff Editor

Journal Requirements:

1. In the ethics statement in the Methods, you have specified that verbal consent was obtained. Please provide additional details regarding how this consent was documented and witnessed, and state whether this was approved by the IRB

Reviewers' comments:

Reviewer's Responses to Questions

**Comments to the Author**

1. If the authors have adequately addressed your comments raised in a previous round of review and you feel that this manuscript is now acceptable for publication, you may indicate that here to bypass the “Comments to the Author” section, enter your conflict of interest statement in the “Confidential to Editor” section, and submit your "Accept" recommendation.

Reviewer #1: (No Response)

Reviewer #2: All comments have been addressed

2. Does this manuscript meet PLOS Global Public Health’s publication criteria? Is the manuscript technically sound, and do the data support the conclusions? The manuscript must describe methodologically and ethically rigorous research with conclusions that are appropriately drawn based on the data presented.

Reviewer #1: Partly

Reviewer #2: Yes

3. Has the statistical analysis been performed appropriately and rigorously?

Reviewer #1: N/A

Reviewer #2: N/A

4. Have the authors made all data underlying the findings in their manuscript fully available (please refer to the Data Availability Statement at the start of the manuscript PDF file)?

Reviewer #1: Yes

Reviewer #2: No

5. Is the manuscript presented in an intelligible fashion and written in standard English?

Reviewer #1: Yes

Reviewer #2: Yes

6. Review Comments to the Author

Reviewer #1: see attached file- revisions are required. Data availability for qualitative research is complex and I do not feel it is required as there are quotations in the manuscript

Reviewer #2: (No Response)

7. PLOS authors have the option to publish the peer review history of their article (what does this mean?). If published, this will include your full peer review and any attached files.

**Do you want your identity to be public for this peer review?** For information about this choice, including consent withdrawal, please see our Privacy Policy.

Reviewer #1: No

Reviewer #2: **Yes: **Simon P.S. Kibira

---

## [Decision Letter · Decision Letter 3]

25 Aug 2023

“Why take the patient back home?”: exploring the lived experiences of caregivers of COVID-19 infected individuals in Blantyre, Malawi.

PGPH-D-22-02004R3

Dear Assoc. Prof Nyondo-Mipando,

We are pleased to inform you that your manuscript '“Why take the patient back home?”: exploring the lived experiences of caregivers of COVID-19 infected individuals in Blantyre, Malawi.' has been provisionally accepted for publication in PLOS Global Public Health.

Best regards,

Julián Alfredo Fernández-Niño, M.D, MPH, MSc, PhD.

Academic Editor

Reviewer Comments (if any, and for reference):

Reviewer's Responses to Questions

**Comments to the Author**

1. If the authors have adequately addressed your comments raised in a previous round of review and you feel that this manuscript is now acceptable for publication, you may indicate that here to bypass the “Comments to the Author” section, enter your conflict of interest statement in the “Confidential to Editor” section, and submit your "Accept" recommendation.

Reviewer #1: All comments have been addressed

Reviewer #2: All comments have been addressed

2. Does this manuscript meet PLOS Global Public Health’s publication criteria? Is the manuscript technically sound, and do the data support the conclusions? The manuscript must describe methodologically and ethically rigorous research with conclusions that are appropriately drawn based on the data presented.

Reviewer #1: Yes

Reviewer #2: Yes

3. Has the statistical analysis been performed appropriately and rigorously?

Reviewer #1: N/A

Reviewer #2: N/A

4. Have the authors made all data underlying the findings in their manuscript fully available (please refer to the Data Availability Statement at the start of the manuscript PDF file)?

Reviewer #1: Yes

Reviewer #2: No

5. Is the manuscript presented in an intelligible fashion and written in standard English?

Reviewer #1: Yes

Reviewer #2: Yes

6. Review Comments to the Author

Reviewer #1: (No Response)

Reviewer #2: No further comments

7. PLOS authors have the option to publish the peer review history of their article (what does this mean?). If published, this will include your full peer review and any attached files.

**Do you want your identity to be public for this peer review?** For information about this choice, including consent withdrawal, please see our Privacy Policy.

Reviewer #1: No

Reviewer #2: **Yes: **Simon Kibira
